# Integrated Transcriptome and Metabolome Analysis of Rice Leaves Response to High Saline–Alkali Stress

**DOI:** 10.3390/ijms24044062

**Published:** 2023-02-17

**Authors:** Guangtao Qian, Mingyu Wang, Xiaoting Wang, Kai Liu, Ying Li, Yuanyuan Bu, Lixin Li

**Affiliations:** 1Key Laboratory of Saline-alkali Vegetation Ecology Restoration, Ministry of Education, College of Life Sciences, Northeast Forestry University, Harbin 150040, China; 2Heilongjiang Academy of Agricultural Sciences, Harbin 150040, China; 3Northeast Branch of National Center of Technology Innovation for Saline-Alkali Tolerant Rice, Daqing 163411, China

**Keywords:** saline–alkali stress, rice, transcriptomics and metabolomics, glutathione metabolism, TCA cycle, linoleic acid metabolism

## Abstract

Rice (*Oryza sativa*) is one of the most important crops grown worldwide, and saline–alkali stress seriously affects the yield and quality of rice. It is imperative to elucidate the molecular mechanisms underlying rice response to saline–alkali stress. In this study, we conducted an integrated analysis of the transcriptome and metabolome to elucidate the effects of long-term saline–alkali stress on rice. High saline–alkali stress (pH > 9.5) induced significant changes in gene expression and metabolites, including 9347 differentially expressed genes (DEGs) and 693 differentially accumulated metabolites (DAMs). Among the DAMs, lipids and amino acids accumulation were greatly enhanced. The pathways of the ABC transporter, amino acid biosynthesis and metabolism, glyoxylate and dicarboxylate metabolism, glutathione metabolism, TCA cycle, and linoleic acid metabolism, etc., were significantly enriched with DEGs and DAMs. These results suggest that the metabolites and pathways play important roles in rice’s response to high saline–alkali stress. Our study deepens the understanding of mechanisms response to saline–alkali stress and provides references for molecular design breeding of saline–alkali resistant rice.

## 1. Introduction

The increasing soil saline-alkalization causes the deterioration of ecological environments and influences crop yields. The total saline–alkali land in China is about 9.91 × 10^7^ ha [1]. The Northeast saline–alkali land in China, e.g., the Songnen Plain, is mainly made up of a high concentration of carbonates, including Na_2_CO_3_ and NaHCO_3_, etc. [2,3]. Therefore, this kind of land is also called Soda saline–alkali land. According to the salt contents and pH value, saline–alkali degrees were classified as mild (salt content < 0.3%, pH 7.1–8.5), moderate (salt content 0.3–0.6%, pH 8.5–9.5), and severe (salt content > 0.6%, pH > 9.5) [4].

Salt stress causes osmotic stress and ion toxicity which inhibit plant growth and development by disrupting the homeostasis of inorganic ions and metabolism of essential organic ions [5,6]. The impact of salt stress on plant is multi-aspect, such as the disorder of metabolic process, inhibition of root system development, and impair of photosynthesis, etc., which lead to a significant decline in crop yields [7,8]. In addition to osmotic stress and ion toxicity, alkali stress causes nutrient deprivation such as the unavailability of phosphorus and irons due to the precipitation of nutrient ions in soil [9,10]. Therefore, alkali stress aggravates the damage to cell structure and activities. The mild alkaline conditions slightly inhibit the growth rate by slowing down metabolic processes (e.g., carbohydrate degradation and nitrogen metabolism retardation). However, moderate and severe alkaline conditions seriously inhibit plant growth and development, and lead to a significant increase in the content of carbohydrate, ROS, and MDA [11]. It has been demonstrated that NaHCO_3_ induces synthesis of large amounts of organic acids [12], such as succinate, citrate, and malate, to compensate for the deficiency of inorganic anions [13], emphasizing the important roles of organic acids in the maintenance of pH and ionic homeostasis. It is also reported that H^+^-ATPases are essential for organic acid secretion in roots under NaHCO_3_ stress [14]. Many studies have clarified that combined saline–alkali stresses cause severer oxidative stress, ionic imbalance, and disrupted osmotic adjustment [15,16,17]. To re-establish osmotic and ionic homeostasis for adaption to the environment, plant cells rapidly accumulate inorganic ions [18,19] and small organic molecules such as betaine, proline, polyamines, polyols, and sugars [20]. For instance, in oat saline–alkali tolerant species, enhanced energy metabolism and increased abundance of organic acids, proline, and betaine facilitate effective osmotic adjustment and cell growth under saline-alkali stress [19]. The ion channels contribute to plant salt tolerance. For example, rice ion transporters, such as OsNHX, OsHKT, and OsSOS1, are particularly important for maintaining the ion homeostasis in rice cells, especially the balance of Na^+^/K^+^. Under salt stress, these ion channel proteins are synthesized in large quantities to facilitate the cells to rebuild the ion homeostasis [21,22,23,24]. On the other hand, rapid accumulation of proline in rice leaves helps to adjust the osmotic balance, thereby increasing the salt tolerance of rice [25].

Saline–alkali stress induces an enhanced production of reactive oxygen species (ROS), e.g., hydrogen peroxide (H_2_O_2_), superoxide (O_2_^−^), and hydroxyl radicals (OH^−^). Salt stress causes an increase in ROS accumulation which disrupts cell activities and leads to metabolic disorder [26]. The ROS scavenging systems include enzymic and nonenzymic antioxidants which cooperate to alleviate the ROS damage to plant cells. The main antioxidant enzymes include peroxidase (POD), superoxide dismutase (SOD), catalase (CAT), ascorbic acid peroxidase (APX), glutathione peroxidase (GPX), and glutathione reductase (GR). SOD bears the brunt in the plant antioxidant system and converts the ROS into H_2_O_2_ and oxygen, and POD, APX, and CAT subsequently convert H_2_O_2_ into oxygen and water. Moreover, to protect membrane integrity, these enzymes coordinately remove MDA generated from lipid peroxidation. Nonenzymic antioxidants include glutathione (GSH), ascorbic acid (ASA), flavonoids, anthocyanins, etc., which fine-tune ROS homeostasis [27]. Saline–alkali stress enhances ASA and GSH abundance which is crucial for maintaining membrane integrity via preventing membrane lipid peroxidation. ASA, GSH, and related antioxidases form a cyclical system to effectively remove free radicals, hence enhancing the antioxidant capacity [27]. An increase in the GSH content enhances plant alkali tolerance [28].

Rice is one of the most important staple crops worldwide. With the growth in the population, the demand for rice is also increasing [29]. Soil salinization and alkalization seriously reduce rice yields and threaten food security. The studies of saline–alkali stress response mechanisms in rice will provide theoretical support for the generation of saline-alkali tolerant rice germplasm. Rice is a salt-sensitive crop, and its salt stress tolerance is related to developmental stages and germplasm [30]. Saline–alkali stress induces saline–alkali changes in soluble sugar contents and antioxidase activities and affects the expression of genes related to the biosynthesis of lignin and wax in rice leaves [31]. The cell wall polysaccharides may also play an important role in improving stress tolerance in a saline–alkali-tolerant rice cultivar [32]. Up to now, there have been many studies on the mechanisms underlying salt stress response in rice, and the mechanisms have been elucidated in depth. However, the mechanisms underlying the saline–alkali stress response in rice have not been fully clarified, and most of the studies are about short-term stress. In this study, we conducted integrated transcriptome and metabolism analyses in rice grown in saline–alkali paddy fields. Our findings indicate that high saline–alkali stress induced significant changes in gene expression and metabolites in rice leaves, including 9347 differentially expressed genes (DEGs) and 693 differentially accumulated metabolites (DAMs). Among the DAMs, amino acids and lipids were accumulated in a large amount. The pathways of the ABC transporter, amino acid biosynthesis and metabolism, glyoxylate and dicarboxylate metabolism, glutathione metabolism, TCA cycle, and linoleic acid metabolism, etc., were significantly enriched with DEGs and DAMs, suggesting that these pathways play important roles in rice’s response to high saline–alkali stress. Our research provides insights into the response of regulatory pathways and important genes in rice to long-term saline–alkali stress. 

## 2. Results

### 2.1. High Saline–Alkali Stress Inhibited Rice Growth and Development

The rice cultivar ‘Tongxi926’ (*Oryza sativa*), widely planted in the northeast area of China, has a certain saline–alkali tolerance (productive under mild saline–alkali conditions). The test paddle field was with pH 9.64 and 1.24 g/kg of salt contents (Table 1), which belonged to severe alkali but mild salt conditions, therefore, we designated it as a high saline–alkali (HSA) condition. Another paddle field was with pH 8.41 and 0.29 g/kg of salt content, which belonged to a mild saline–alkali (MSA) condition. Since Tongxi926 grew well under this condition, we used it as a control condition.

The rice (*Oryza sativa*) growth and development were severely inhibited in HSA fields, with a dwarfed, reduced tillering number and many reddish brown leaves (Figure 1A,B and Appendix A). Under HSA stress, the MDA content increased significantly (Figure 1C), suggesting that HSA stress induced excessive ROS production, leading to serious membrane lipid peroxidation. In response, the activities of antioxidases, such as SOD, POD, APX, and CAT, as well as the proline content in rice leaves, increased sharply (Figure 1D,E), indicating that HSA and ROS stress strongly activated the ROS scavenging and osmotic adjustment systems which maintain membrane integrity and cell activities.

### 2.2. Transcriptomic Analysis of Rice Leaves

To uncover the transcriptional gene responses to saline–alkali stress, the transcriptome analysis of rice leaves grown under MSA and HSA conditions was conducted. A total of 99.69 million raw reads were generated. After filtering, 98.34 Mb of clean reads were obtained, with a Q30 of >95%, and GC content ranging from 49.59% to 50.27% (Table 2). These results confirmed the high quality of the assembled transcripts.

### 2.3. Identification of DEGs in Response to High Saline–Alkali Stress 

The principal component analysis (PCA) revealed significant differences between the MSA and HSA groups (Figure 2A). A total of 9347 differentially expressed genes (DEGs) were identified using DESeq software, with 5087 upregulated and 4260 downregulated DEGs for HSA vs. MSA (Figure 2B). To clarify the molecular mechanisms underlying the saline–alkali stress response in rice leaves, KEGG and gene ontology (GO) enrichment analyses were performed. KEGG analysis indicates that the DEGs were enriched in 124 pathways. The TOP20 were related to a plant–pathogen interaction (dosa04626), peroxisome (dosa04146), plant hormone signal transduction (dosa04075), alpha-linolenic acid metabolism (dosa00592), linoleic acid metabolism (dosa00591), glutathione metabolism (dosa00480), amino acid metabolism (dosa00400, dosa00360, dosa00280, dosa00250), and fatty acid degradation (dosa00071), etc. (Figure 2C). 

The total DEGs were subjected to GO analysis to achieve a broader functional characterization. As a result, the DEGs were categorized into fifty-two subcategories within three main categories: twenty-three subcategories in Biological Processes (BP), thirteen in Cellular Components (CC), and sixteen in Molecular Functions (MF) (Figure 3A). In the BP category, the TOP3 enriched pathways were the cellular process (GO:0009987, 3464 genes), metabolic process (GO:0008152, 2951 genes) and single-organism process (GO:0044699, 2662 genes). In the CC category, the TOP3 enriched pathways were the cell (GO:0005623, 4388 genes), cell part (GO:0044464, 4378 genes), and organelle (GO:0043226, 3311 genes). Additionally, in the MF category, the TOP3 enriched pathways were the binding (GO:0005488, 3369 genes), catalytic activity (GO:0003824, 3064 genes), and transporter activity (GO:0005215, 547 genes) (*p*-value < 0.05) (Figure 3A). In MF, the most significant enrichment of DEGs were in the catalytic activity (GO:0003824), cofactor binding (GO:0048037), and chitinase activity (GO:0004568) (Figure 3B). In BP, the most significant enrichment of DEGs were in response to stimulus (GO:0050896), response to stress (GO:00069505), the small molecule metabolic process (GO:0044281), and organic acid metabolic process (GO:0006082) (Figure 3C). In CC, the most significant enrichment of DEGs were in plastid (GO:0009536) and chloroplast (GO:0009507) (Figure 3D). These results indicate that the redox, defense, and organic metabolism processes were the most activated under high saline–alkali stress.

### 2.4. Transcription Factors Related to High Saline–Alkali Stress

During signal transduction in response to stress, transcription factors (TFs) function as a bridge between stimulus signals and downstream genes. They transmit signals and regulate the expression of target genes by binding to the cis-regulatory elements in the promoter. Transcriptome analyses of the rice leaves indicate that the expression levels of some TFs were significantly modified for HSA vs. MSA. There were 500 differentially expressed TFs (DETFs) identified, of which 321 were upregulated and 179 were downregulated. The DETFs were classified into 57 TF families. Most of the DETFs were enriched in bHLH, ERF, MYB, NAC, C2H2, WRKY, and bZIP families (Figure 4A, red arrows), and most of the TF targets were under the regulation of ERF, Dof, and BBR-BPC families (Figure 4B, blue arrows). Notably, most of the DETFs were upregulated in the major families of ERF, MYB, NAC, and WRKY families, suggesting that these TFs participated in the signal transduction in rice’s response to high saline–alkali stress.

Among these TFs, we checked the expression of some TFs and their target genes, such as *DOF11* and target *SWEET14* [33], and *WRKY53* and target *SWEET2a* [34]. The expression trends were the same as those of the transcriptomes (Appendix A). Moreover, Pre-mRNA alternative splicing (AS) frequencies of DEGs in HSA vs. MSA had significant differences, such as XMIR, XIR, XAE, MIR, and IR, which differed by 1.5 (IR) to 3.4 times (XMIR) (Appendix A). AS is the most significant mechanism for producing multiple protein isoforms from a single gene. This co-/post-transcriptional mechanism is highly induced by abiotic stress and involves a large number of stress-related genes [35]. The alteration of AS in HSA leaves may improve stress tolerance in rice.

### 2.5. Non-Targeted Metabolomic Analysis of Differentially Accumulated Metabolites

To further illustrate the changes in metabolites in rice leaves triggered by HSA stress, the untargeted metabolome was profiled using HPLC-ESI-MS/MS. The PCA analysis revealed significant differences of metabolites between the MSA and HSA groups (Figure 5A). Additionally, the orthogonal partial least squares discriminant analysis (OPLS-DA) indicated high reproducibility of the results and enabled further differential metabolite analysis (Figure 5B). In total ion chromatograms (TIC), 7261 metabolites were detected, of which 4065 were positive and 3196 were negative in the ion mode, respectively (Appendix A). Among these, a total of 693 differentially accumulated metabolites (DAMs) were identified for has vs. MSA, of which 430 were upregulated and 263 were downregulated (Figure 5C). A KEGG enrichment analysis revealed that the DAMs were enriched in twenty-three metabolic pathways (Appendix A), including eleven amino acid metabolic pathways (blue arrows in Figure 5D), five organic acid metabolic pathways (blue lines in Figure 5D), ABC transporters, and carbon metabolism, etc. The TOP5 enriched pathways were the biosynthesis of amino acids, alanine, aspartate and glutamate metabolism, aminoacyl-tRNA biosynthesis, and the ABC transporters and citrate cycle (TCA cycle) (Figure 5D).

To adapt to high saline–alkali stress, the rice leaves accumulated a large amount of lipids (250), organic acids (93), organic oxygen compounds (73), phenylpropanoids, and polyketides (59) (Appendix A). The two hundred fifty kinds of lipids contained eighty fatty acyls (including twenty fatty acids, ten octadecanoids, nine eicosanoids, seven lineolic acids, seven fatty amides, and seven fatty acyl glycosides, etc.), Forty-six prenol lipids, forty-four glycerophospholipids/nine glycerolipids, thirty-seven polyketides (include thirty-five flavonoids) and sixteen steroids, etc. (Appendix A, Appendix A). Additionally, the most increased lipids were fevicordinB2-gentiobioside [steroids, log_2_FC (Fold Change) = 17.18], all-trans-3,4-didehydroretinoate (prenol lipids, log_2_FC = 7.92), N-butanoyl-lhomoserine lactone (fatty acyls, log_2_FC = 7.89), DG (16:0/20:5/0:0) (glycerolipids, log_2_FC = 6.78), and luteolin 7-[6′′-(2-methylbutyryl) glucoside] (flavonoids, log_2_FC = 6.11). Additionally, the most decreased lipids were asprellic acid C (prenol lipids, log_2_FC = −24.96), goyaglycoside C (steroids, log_2_FC = −7.49), asparagoside C (steroids, log_2_FC = −5.65), and 12-Oleanene-pentol-22-angeloyloxy-23-al (prenol lipids) with log_2_FC = −5.28 (Appendix A). It is worth mentioning that among 16 Steroids, 14 were reduced; among 46 prenol lipids, 35 were reduced; and among 35 flavonoids, 12 were reduced, suggesting that they might be consumed during stress adaption (Appendix A).

The 93 kinds of organic acids included 85 carboxylic acids which contained 79 amino acids, and the abundance of most amino acids were elevated (Appendix A). The most increased ones were threoninyl-proline (log_2_FC = 25.85), 2-Pentanamido-3-phenylpropanoic acid (log_2_FC = 25.42), isovalerylglutamic acid (log_2_FC = 8.59), ustiloxin B (log_2_FC = 7.19), and tetragastrin (log_2_FC = 6.88). Additionally, the most decreased ones were glutathionate (1-) (log_2_FC = −2.68), 20-COOH-leukotriene E4 (log_2_FC = −1.44), and L-acetopine (log_2_FC = −1.11), N(6) − (octanoyl)lysine (log_2_FC = −0.94), and arginyl-serine (log_2_FC = −0.71) (Appendix A). These results imply that high saline–alkali stress activated biosynthesis and the metabolic processes of these compounds and could promote plants to counteract harsh environment.

### 2.6. Integrative Analysis of Transcriptome and Metabolome

To clarify the regulation network for HSA response, integrated transcriptomic and metabolomic analyses was conducted. KEGG analysis revealed that DEGs and DAMs were enriched in 53 pathways, (Appendix A), among which 11 pathways were related to amino acid metabolism and biosynthesis (Figure 6, blue arrows). Additionally, the most enriched pathways were ABC transporters, alanine, aspartate and glutamate metabolism, aminoacyl-tRNA biosynthesis, phenylalanine metabolism, and glutathione metabolism. In addition, glyoxylate and dicarboxylate metabolism, the citrate cycle, and tryptophan metabolism were also well enriched (Figure 6), suggesting that these metabolism pathways were enhanced in response to high saline–alkali stress.

#### *2.6.1. Glutathione Metabolism May Enhance ROS Scavenging under High Saline–Alkali Stress* 

In plants, an initial stress response is generally related to glutathione metabolism. Reduced glutathione (GSH) acts as a redox buffer to maintain the intracellular reduction status [36]. The AsA–GSH cycle is indispensable in eliminating H_2_O_2_. In the cycle, GSH acts as a reducer to transform ascorbate from its oxidized form (dehydroascorbate, DHA) to its reduced form (AsA) [37]. GR reduces oxidized glutathione (GSSG) into reduced glutathione. GSH could be modified via glutathione-S-transferase (GST) and glutathione peroxidase (GPX) to scavenge H_2_O_2_ by catalyzing the formation of GSH conjugates [38].

In Tongxi926 leaves under HSA stress, four metabolites and sixty-four genes were involved in glutathione metabolism (Figure 7A). GSSG (glutathione disulfide), L-glutamate, L-gamma-glutamyl-L-amino acid, and 5-oxoproline were found to be significantly accumulated. On the other hand, the transcription levels of most related genes were upregulated. For example, two *GR* (glutathione reductase), three *GPX* (glutathione peroxidase), and 14 *GST* (glutathione s-transferase) genes were upregulated, while two *GST* and two *G6PDH* (glucose-6-phosphate 1-dehydrogenase) genes were downregulated. Enhanced GSSG accumulation and *GR* and *GPX* expression should facilitate the GSSG–GSH cycle. Although the content of GSH did not change substantially, the content of oxidized GSH (GSSG), GSH conjugates, 5-L-glutamyl-L-amino acid, and L-glutamate, elevated significantly (Figure 7A). It has been demonstrated that glutamate acts as a hub which can link amino acid metabolism to organic acids via the TCA cycle [39], and can produce proline in arginine and proline metabolism. In Tongxi926 leaves under HSA stress, enhanced L-glutamate accumulation might lead to a significant increase in proline (Figure 7A) and its derivatives, e.g., threoninyl-proline, one of the most increased metabolites (log_2_FC = 25.85) (Appendix A). 

We confirmed the expression of some related genes using RT-qPCR. 6-GPDH (6-phosphogluconate dehydrogenase) is involved in glutathione content maintenance and ROS scavenging by utilizing NADPH. GGT (Gamma-glutamyl transferase) is the only enzyme capable of degrading glutathione in extracellular spaces and vacuoles [40]. RT-qPCR determination indicates that the expression of *6-PGDH* (*Os11g0484500*), *GGT* (*Os01g0151500*), *GPX* (*Os02g0664000*) and *GST* (*Os10g0525800*) was significantly altered, and the change trends were identical to those in transcriptomes (Figure 7B). The changes in these genes imply the important role of glutathione metabolism in scavenging ROS in rice leaves under HSA stress.

#### *2.6.2. TCA Cycle May Provide Energy to Improve Rice Stress Tolerance* 

Under alkali stress, many pathways such as α-oxidation, glycolysis, and the tricarboxylic acid (TCA) cycle are enhanced to provide the energy to improve alkali tolerance [41]. It is also reported that under alkali and salt stress, the plant TCA cycle produces more energy and accelerates the physiological and metabolic response to stress [42]. In Tongxi926 leaves under the HSA condition, glyoxylate and dicarboxylate metabolism, and TCA cycle were likely the major routes for supplying energy. There were four DAMs and 23 DEGs enriched in the TCA cycle. An abundance of citrate, (S)-malate, succinate, and 2-oxoglutarate demonstrated that it changed significantly (Figure 8A). For citrate biosynthesis, PDH E1α (Pyruvate dehydrogenase E1 α-subunit) catalyzes oxidative decarboxylation of pyruvate to acetyl-CoA, which produces citrate [43]. Upregulated expression of *PDH E1α* should enhance the conversion of pyruvate into citrate, whose content was significantly elevated (Figure 8A). FH (fumarate hydratase) can reversibly convert (S)-malate into fumarate, and SDH (succinate dehydrogenase) subsequently reversibly converts fumarate into succinate. Upregulated expression of *FH* and *SDH* promotes the conversion between (S)-malate and succinate. The reduction of (S)-malate might be due to the conversion into succinate and/or citrate, which were increased significantly (Figure 8A). DLST (dihydrolipoamide succinyltransferase), IDH (Isocitrate dehydrogenase), and IMDH (isopropylmalate dehydrogenase) catalyze 2-oxoglutarate to succinate or isocitrate which is subsequently converted into citrate. Upregulated expression of *DLST*, *IDH*, and *IMDH* might accelerate 2-oxoglutarate consumption and succinate and citrate accumulation (Figure 8A). RT-qPCR detection confirmed that the change trends of these gene expression levels were consistent with those in the transcriptome (Figure 8B). The alteration of these metabolites and related genes might be advantageous for leaf cells to produce more energy to counteract HSA stress.

#### *2.6.3. Linoleic Acid Metabolism May Alleviate the Membrane Damage Caused by High Saline–Alkali Stress* 

Linoleic acid (LA) is a polyunsaturated fatty acid essential for eukaryotes development [44]. In plants, a higher LA content is conducive to maintaining the integrity and fluidity of the cell membrane, which is beneficial for plant adaptation to environmental stresses [45,46]. The linoleic acid derivatives, linoleate and alpha-linolenate, are the two most common polyunsaturated fatty acids in plants [47]. Lipoxygenase functions in responses to osmotic stress [48]. Linoleate 9S-lipoxygenases and lipoxygenases (LOXs) oxidize linoleate to form 9/13(S)-HPODE, respectively. 9-HPODE can induce GSH oxidation independent of secondary products or H_2_O_2_. 13-OxoODE is produced from 13-HODE by a NAD+-dependent dehydrogenase (Figure 9A). In Tongxi926 leaves under HSA stress, there were three DAMs and 11DEGs were enriched in the linoleic acid metabolism (Figure 9A). 13-OxoODE and 9(S)-HODE were accumulated in a high level, while 12,13-DHOME reduced significantly; both of them may affect membrane function by altering the membrane structure and fluidity. RT-qPCR determination of the LOX genes validated the changes in the gene expression levels, identifying them as consistent with those in the transcriptome (Figure 9B). The DAMs and DEGs may facilitate to reduce membrane damage under HSA stress.

## 3. Discussion

Plant growth and development are restricted by various environmental stresses, such as drought, temperature, salinity, and alkalinity. Salt stress can cause membrane lipid peroxidation, ruin cell membrane permeability, and consequently affect plant growth and development. In addition to these damages, alkali stress can also cause high pH stress, damage the rhizosphere structure, and block nutrient absorption [4]. In this study, a saline–alkali-tolerant rice cultivar ‘Tongxi926’ was selected for transcriptome and metabolome analysis. The pH value of the test paddy field was higher than 9.5, but the salt content was lower than 0.3%, implying that alkali stress in rice was more serious than salt stress.

Oxidative stress induced by saline–alkali may damage the structure of cell membranes, and SOD and POD have important roles in preventing cell membrane damage. In our study, the antioxidant activity of ROS, POD, APX, and CAT enzymes under HSA were higher than MSA, while proline accumulates more in rice leaves under HSA stress (Figure 1C,D). This may be because saline–alkali stress can quickly produce a large amount of ROS to damage plant cells, so reducing the accumulation of ROS is one of the adaptive mechanisms for plants to improve their resistance to saline–alkali stress [49,50].

In plants, the protection mechanisms against saline–alkali stress involve metal sequestration and ROS scavenging systems [51]. Glutathione metabolism is an essential pathway in defense machinery. Glutathione acts as an antioxidative substrate by conjugating with toxic electrophilic compounds, scavenging free radicals, and reducing peroxides [52,53]. For example, a salt-tolerant rice variety has a higher glutathione content than a salt-sensitive variety [36]. In plants exposed to salt stress, the glutathione redox state is disturbed and GSH is oxidized to GSSG [54]. Increased H_2_O_2_ accumulation causes a decrease in the GSH/GSSG ratio, and the equilibrium between GSH and NADPH redox couples is broken [55]. In Tongxi926 leaves under HSA stress, the oxidized glutathione (GSSG) increased 3.5-fold (Figure 7A; Appendix A). Although the glutathione content did not alter, the amount of two products, 5-L-Glutamyl-L-amino acid and L-Glutamate, increased significantly (Figure 7A), indicating that GSH was consumed to convert into the two products and its oxidation product GSSG. On the other hand, two *GR* genes, three *GPX* genes, one *6-PGDH* gene, and one *IDH* gene were upregulated, while two G6PDH genes were downregulated (Figure 7A). The significant changes in the expression of these genes, together with the increase in the GSSG content, may accelerate the GSH-GSSG cycle, which is conducive to ROS scavenging. Moreover, glutamate has been demonstrated to act as a hub which can link amino acid metabolism to organic acids via the TCA cycle [39] and can produce proline in arginine and proline metabolism. Enhanced L-glutamate accumulation may lead to a significant increase in the content of proline (Figure 6A) and its derivatives, including threoninyl-proline, one of the most increased metabolites (log_2_FC = 25.85) (Appendix A). The above changes indicate that these genes and metabolites are essential for rice resistance to high saline–alkali stress.

Some studies have shown that GSH regulates plant stress tolerance by affecting the expression of genes involved in the plant hormone signal transduction and plant–pathogen interaction pathways [56,57]. Since a non-targeted metabolomics approach based on HPLC-ESI-MS/MS was employed in this study, no related metabolites were determined in these two pathways. Therefore, we used the STRING database (https://cn.string-db.org/, accessed on 6 February 2023) to perform protein–protein interaction prediction between glutathione metabolism and these two pathways. Our results show that glutathione metabolism was closely linked to plant hormone signal transduction and plant–pathogen interaction, respectively (Appendix A). Specifically, APX1, a member of the APX family which is the key enzyme of the H_2_O_2_-detoxification system and an important component of the glutathione cycle in plants, associates with PP2C30 and PP2C06 (Appendix A, red arrows), the protein phosphatases which negatively regulate ABA signaling but positively regulate abiotic stress signaling [58], suggesting a connection between glutathione metabolism and the ABA signaling pathway in the regulation of rice tolerance to saline–alkali stress. Moreover, APX1 also associates with CERK1 (chitin elicitor receptor kinase 1), CML4 (calmodulin-like4), CML5, CML11, and CML14, the factors in the plant–pathogen interaction pathway (Appendix A, red arrows). CERK1, a key receptor in plant–microbe interactions [59], is demonstrated to be also involved in salt tolerance regulation in plants [60]. CML proteins act as Ca^2+^ sensors and improve rice tolerance to salt stress [61,62,63,64,65]. The association of APX1 with these proteins linked glutathione metabolism to the plant–pathogen interaction pathway in response to high saline–alkali stress. In conclusion, APX1 acts as a hub that connects glutathione metabolism and other pathways and enables these pathways to jointly regulate rice’s response to saline–alkali stress.

A transcriptome analysis of the salt stress response mechanism in wild rice (Dongxiang) shows 6867 DEGs in leaves for salt stress vs. normal conditions. These DEGs distribute in multiple regulatory pathways, such as the biosynthesis of secondary metabolites, carbohydrate metabolism, metabolism of terpenoids and polyketides, translation, and transport and catabolism, etc. [66]. Compared with TongXi926 under high saline–alkali stress, wild rice exhibits similarities and differences in stress response mechanisms. Specifically, the DEGs in the TF family were relatively similar in two rice species. For example, the TFs are enriched in the bHLH, MYB, bZIP, ERF, and NAC families, and the two rice species share WARK24 (Os05g0343400), WRKY28 (Os06g0649000), and WRKY70 (Os05g0322900), etc. This is consistent with the conclusion from some other studies that WRKY-type TFs play an important role in the adaptation to abiotic stresses [67,68,69]. On the other hand, the DEGs in wild rice under salt stress are mainly enriched in the biosynthesis of secondary metabolites, amino sugar and nucleotide sugar metabolism, RNA transport, and mRNA surveillance pathways, etc., while the DEGs in TongXi 926 under HSA stress are most enriched in the plant–pathogen interaction, plant hormone signal transduction, peroxisome, and glutathione metabolism, etc. (Figure 2C), indicating that different rice varieties have distinct points in stress response mechanisms, which is also related to the different stress types.

Under saline–alkali stress, photosynthesis, respiration, hormone action, and nutrition metabolism are disrupted [70]. To adapt to saline–alkali stress, plants accumulate numerous small-molecule organic solutes such as carbohydrates, polyphenols, amino acids, and other organic acids [71]. An abundance of amino acids are enhanced to function as effective antioxidants to scavenge free radicals in plants [72]. Amino acids are involved in various biochemical processes, protein biosynthesis, and signaling in stress response [73,74]. For example, tyrosine functions as a hub connecting many metabolic pathways, and is also the precursor of special metabolites such as non-protein amino acids, attractants, and defense compounds [75,76]. Proline is the most well-known compatible solute and plays a crucial role in osmoregulation and cell membrane stability to improve plant tolerance to abiotic stress [77,78,79]. In Tongxi926 leaves under HSA stress, the abundance of amino acids was sharply increased, such as with proline and its derivatives (threoninyl-proline, L-proline, L-glycyl-L-hydroxyproline, leucylproline, aspartyl-L-proline, and phenylalanylproline) (Appendix A). Moreover, tyrosine biosynthesis and the content of tyrosine derivatives were enhanced (Figure 6; Appendix A). These results imply the critical roles of amino acids in improving rice’s tolerance to high saline–alkali stress. 

Another thing worth noting is that the rice under HSA stress had many reddish-brown leaves (Figure 1B), which looked like a physiological disease, and which is rarely reported at present. When the soil has high salinity and alkalinity, especially when pH > 8.5, many CO_3_^2−^, HCO_3_^−^, and Na^+^ ions greatly reduce the effective utilization of trace elements and cause nutrient imbalance such as phosphorus, potassium, and zinc deficiency, and result in physiological diseases. In the transcriptome of Tongxi926 leaves under HSA stress, there were many disease-resistance-related DEGs, including seventy-six *RGA* (resistance gene analog)1-5 genes, fourteen *LRK10* (leaf rust 10 disease-resistance locus receptor-like protein kinase) and twelve *LRK10-L* (LRK10-Like) genes, seventeen *RPPL1* (resistance to P. pachyrhizi), *RPP13*, *RPP13L2-4* genes, fourteen *RPM1* (resistance to pseudomonas syringae pv. maculicola 1) genes, seven *EDR4* (enhanced disease resistance 4) and *EDR2-L* genes, four *PIKS-1/2* (phosphatidylinositol (PI) kinases) genes, seven *PIK-1/2*, 6 *PIK6-NP*, 5 *PIKM2-TS*, and three *RPS2* (ribosomal protein S2) genes, etc. (Appendix A). Moreover, some TFs and their targets, DOF11, SWEET14, WRKY53, and SWEET2a (Appendix A), have been reported to be involved in sheath blight resistance in rice [33,34]. Moreover, chitinase activity (GO:0004568), one of the most significant enriched processes in MF in Tongxi926 leaves (Figure 3B), has been demonstrated to significantly correlate with sheath blight resistance in McCHIT1-transgenic rice line plants [80]. All these suggest that Tongxi926 grown in the HSA paddy field might have a physiological disease, which roused defense mechanisms.

Since the stress has always existed during the growth and development of rice, changes in metabolites and genes are the result of long-term stress. As it is a field experiment, the natural environmental factors are relatively complex. In addition to the major HSA stress, there may be other factors. However, the influence of various factors is the actual situation often encountered in field production. Our research results have a practical reference value for the problems encountered in actual agricultural production.

## 4. Materials and Methods

### 4.1. Plant Materials and Treatments

The saline–alkali tolerant rice cultivar ‘Tongxi926’ (*O. sativa*) (provided by Heilongjiang Zhongbao Agricultural Technology Development Co., Ltd., Daqing, China) planted at 20 May 2021, in two different fields about two kilometers apart (Zhaoyuan Comt, Heilongjiang Province, 45°53, N, 125°07, E), which belonged to mild saline–alkali (MSA, used as control) and high saline–alkali (HSA) conditions, respectively. Soil physical and chemical properties are shown in Table 1. 

### 4.2. Total RNA Isolation, Library Construction, and Sequencing

The rice leaves at booting stage were harvested for RNA Sequencing. For sampling of control rice leaves, the green leaves were randomly collected; for test rice leaves, the reddish-brown leaves were randomly collected (shown in Figure 1B). Total RNA was extracted using the TRIzol reagent (Invitrogen, Carlsbad, CA, USA) according to the manufacturer’s protocol. RNA purity and integrity were assessed using NanoDrop 2000 spectrophotometer (Thermo Scientific, Waltham, MA, USA) and Agilent 2100 Bioanalyzer (Agilent Technologies, Santa Clara, CA, USA), respectively. Thereafter, mRNA was enriched and purified with oligo (dT) and fragmented using the fragmentation buffer. First- and second-strand cDNA were synthesized from cleaved mRNA fragments by using reverse transcription and random hexamer primers. After purifying these fragments, we washed them with the EB buffer for end reparation of the poly(A) addition and ligated them to the sequencing adapters. The size of the fragment was determined through agarose gel electrophoresis, followed by enriching the library of purified cDNA via PCR amplification. Finally, six libraries were sequenced on the Illumina sequencing platform (HiSeqTM 2500), provided by OE Biotech Co., Ltd. (Shanghai, China).

The libraries were sequenced on an Illumina HiSeq X Ten platform, and 150 bp paired-end reads were generated. For obtaining clean reads, raw fastq data were first processed using Trimmomatic to remove low quality reads. The clean reads were mapped to the rice genome (GRCh38) using HISAT2. According to the length of each gene and the number of reads mapped to this gene, the FPKM was calculated.

### 4.3. Quantitative Real-time PCR Validation

The gene-specific primers were designed using primer3Plus software (http://www.primer3Plus.com/, accessed on 13 October 2022), and the *UBQ5* gene was used as the endogenous control. The RT-qPCR reactions were performed using the CFX96^TM^ real-time system (Bio-Rad, Hercules, CA, USA). To test the specificity of the amplified products, the melting curve was generated and analyzed by increasing the temperature from 50 ℃ to 95 ℃. Subsequently, the relative expression of genes was calculated using the 2^−ΔΔCT^ method. The gene specific primers are summarized in Appendix A. The total RNA used for quantitative real-time PCR validation was consistent with that for RNA sequencing.

### 4.4. Metabolite Extraction 

Freeze-dried leaves were grinded at 60 HZ for 2 min, and the powder of 100 mg extracted using 1.0 mL of 70% aqueous methanol was stored at 4 °C overnight. Samples were centrifuged at 13000 rpm, 4 °C for 15 min, and the supernatants were finally filtered with a 0.22 μm microfilter before LC-MS analysis.

### 4.5. Metabolite Detection

The sample extracts were used to analyze the metabolic profiling using an ACQUITY UHPLC system (Waters Corporation, Milford, CT, USA) coupled with an AB SCIEX Triple TOF 5600 System (AB SCIEX, Framingham, MA, USA). The analytical conditions were as follows: UPLC: column, ACQUITY UPLC BEH C18 column (1.7 μm, 2.1 × 100 mm); mobile pHSAe, eluent (A) (containing 0.1% formic acid, *v*/*v*) and (B) acetonitrile (containing 0.1% formic acid, *v*/*v*). The following gradient program was employed: 0 min, 5% B; 2 min, 20% B; 4 min, 25% B; 9 min, 60% B; 14 min, 100% B; 18 min, 100% B; 18.1 min, 5% B and 19.5 min, 5%B. The flow rate and column oven were set as 0.4 mL/min and 45 ℃, respectively. The injection volume was 1 μL.

Data acquisition was performed in full scan mode (70–1000 m/z) combined with the IDA mode. Parameters of mass spectrometry were as follows: ion source temperature, 550 °C (positive ion mode), and 550 °C (negative ion mode); ion spray voltage, 5.5 KV (positive ion mode) and 4.5 KV (negative ion mode); curtain gas of 35 PSI; declustering potential, 100 V (positive ion mode), and −100 V (negative ion mode); collision energy, 10 eV (positive ion mode), and −10 eV (negative ion mode); and interface heater temperature, 550 °C (positive ion mode), and 600 °C (negative ion mode). 

### 4.6. Functional Enrichment Analysis of DEGs 

Differential expression analysis of two groups was performed using the DESeq2 R package (1.20.0). The significant differential expression genes (DEGs) were defined as *p* value < 0.05 and |log_2_FoldChange| ≥ 1. DEGs were analyzed using hierarchical cluster analysis to determine how they expressed in various groups and samples. Gene ontology database (http://www.geneontology.org/, accessed on 26 October 2022) and KEGG orthology-based annotation system (KOBAS) software were performed to analyze GO (gene ontology) enrichment and KEGG (Kyoto encyclopedia of genes and genomes) pathway enrichment of DEGs, respectively. KEGG pathway and GO enrichment analyses were based on *p*-value < 0.05.

### 4.7. Statistical Analysis 

For metabolite statistics, the positive and negative data were analyzed using a self-built database (Luming Biotech CO., Ltd., Shanghai, China), and other metabolite databases (MassBank, HMDB, Lipidmaps, PubChem, and METLIN). Principle component analysis (PCA) and orthogonal partial least-squares-discriminant analysis (OPLS-DA) were carried out to visualize the metabolic alterations among the experimental groups. The differentially accumulated metabolites (DAMs) were selected on the basis of the variable influence on projection (VIP) values obtained from the OPLS-DA model and *p*-value; metabolites with VIP > 1.0 and *p*-value < 0.05 were considered as DAMs. The metabolite pathways were analyzed in the KEGG database. 

### 4.8. Physiological Measurements

The rice leaves at the booting stage were harvested for enzymic activity determination. The POD activity was determined using guaiacol colorimetry, SOD activity using nitro blue tetrazolium photoreduction, CAT activity using the hydrogen peroxide method, and APX was determined according to the principle that H_2_O_2_ reduces the content of AsA [17]. The MDA content was determined using the thiobarbituric acid method, and the proline content using the acidic ninhydrin colorimetry method [17]. Three biological replicates per sample. Statistical significance was defined via one-way ANOVA, *p* < 0.05.

## 5. Conclusions

In this study, we focused on the important genes and metabolites involved in the long-term saline–alkali stress in rice leaves via transcriptome and metabolome analysis. A total of 693 DAMs and 9347 DEGs were identified for HSA vs. MSA, respectively. The DEGs of transcription factors were mainly enriched in *bHLH*, *ERF*, *MYB*, *NAC*, *C2H2*, *WRKY*, and *bZIP* families, protruding the significance of transcriptional regulation in rice’s response to HSA stress. Under HSA stress, the contents of amino acids and lipids increased most, suggesting their essential roles in improving rice tolerance to saline–alkali stress. The DEGs and DAMs were most significantly enriched in the pathways of amino acid biosynthesis and metabolism, the ABC transporter, glyoxylate and dicarboxylate metabolism, glutathione metabolism, TCA cycle, and linoleic acid metabolism, indicating that these pathways are important for the regulation of rice’s response to saline–alkali stress. 

## Figures and Tables

**Figure 1 ijms-24-04062-f001:**
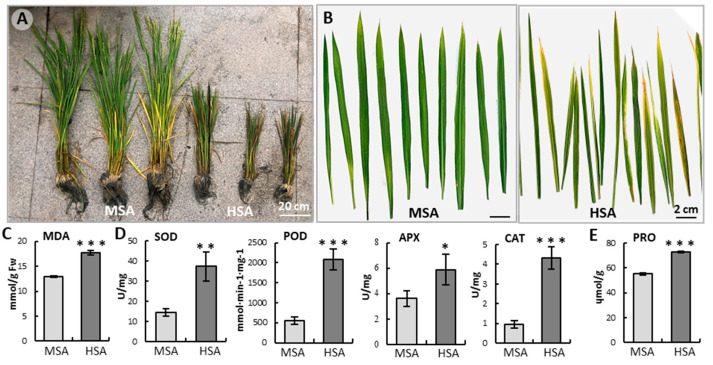
The growth and development of rice was inhibited under high saline–alkali (HSA) conditions. (**A**) The rice plants grown under mild saline–alkali (MSA) and HSA conditions. (**B**) The leaves were severed from plants at the booting stage, and were used for enzyme activity determination, transcriptome, metabolome analyses, and RT-qPCR. (**C**–**E**) MDA content (**C**), antioxidant enzyme activities (**D**), and proline content (**E**) in rice leaves. Data are mean ± SE of three biological replicates. *, *p* < 0.05; **, *p* < 0.01; ***, *p* < 0.001; Student’s t-test. Abbreviations: ascorbate peroxidase (APX); catalase (CAT); malondialdehyde (MDA); peroxidase (POD); proline (PRO); superoxide dismutase (SOD).

**Figure 2 ijms-24-04062-f002:**
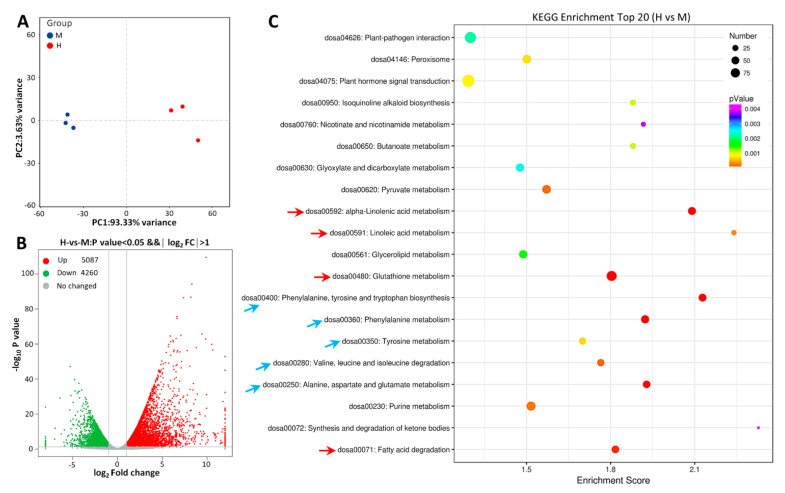
General overview of differentially expressed genes (DEGs). (**A**) PCA diagram. Blue dots, MSA group; red dots, HSA group. X−axis and Y−axis indicate the first and second principal components (PC1 and PC2), respectively. Score plots of PC1 and PC2 show cohesion within groups and separation between the MSA and HSA group, respectively. (**B**) Volcano map of transcriptome genes for HSA vs. MSA. Red dots, upregulated genes; green dots, downregulated genes; gray dots, genes with no significance. (**C**) KEGG enrichment of DEGs in TOP20 pathways. The *p* value is presented in a color scale, the size of the dots represents DEG number mapped in each pathway.

**Figure 3 ijms-24-04062-f003:**
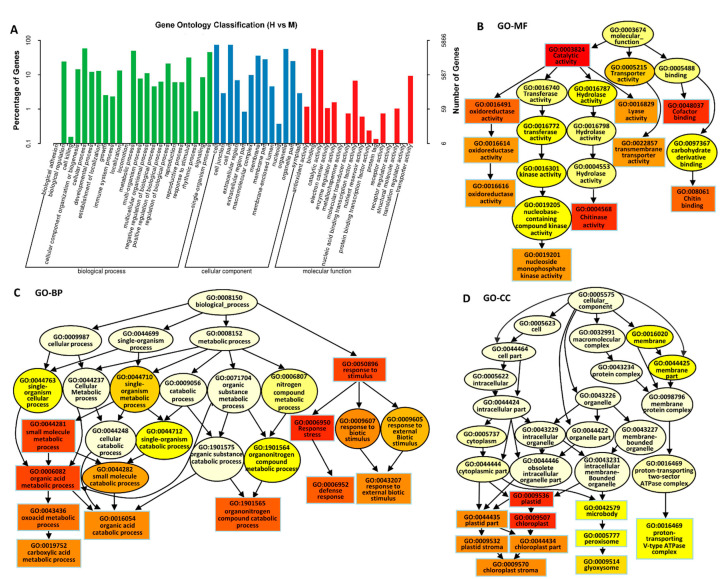
Functional annotation of DEGs based on gene ontology categorization. (**A**) GO enrichment analysis of DEGs. The DEGs were categorized into three main categories, Molecular Functions (MF), Biological Processes (BP), and Cellular Components (CC). (**B**–**D**) The functional categories of DEGs in MF (**B**), BP (**C**), and CC (**D**), respectively. Rectangles/circles contain the GO term number and category name. Rectangle and oval colors represent the relative significances, ranging from red (the most significant, *p* < 0.0001), orange (the second significant, *p* < 0.001), yellow (the third significant; *p* < 0.01) to light yellow (the least significant, *p* < 0.05).

**Figure 4 ijms-24-04062-f004:**
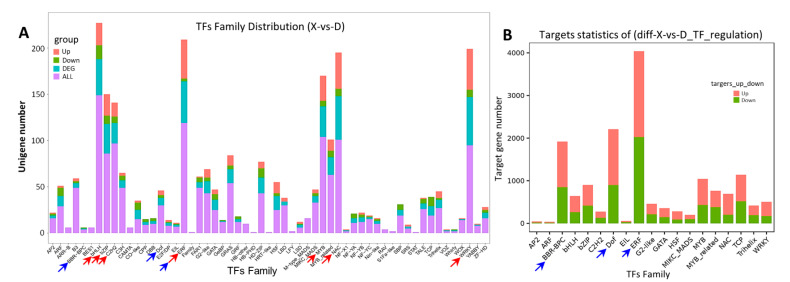
The TF families and their targets. (**A**) Distribution of DEGs in TF families. X-axis and Y-axis represent TF families and gene number, respectively. The red and green bars indicate the number of upregulated and downregulated TFs, respectively. The cyan bars indicate total DEGs, and the purple bars indicate total genes identified in TF families. (**B**) The target gene families of TFs. The red and green bars represent the number of upregulated and downregulated target genes, respectively.

**Figure 5 ijms-24-04062-f005:**
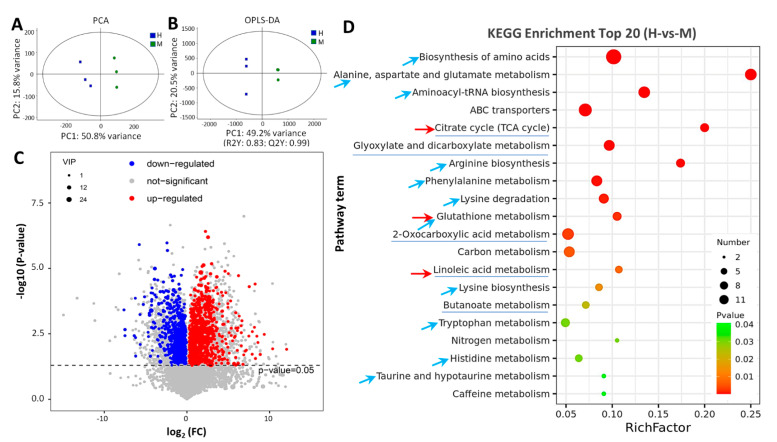
General overview of differentially accumulated metabolites (DAMs) in rice leaves under HSA stress. (**A**) PCA score scatter plots of metabolites determined via UHPLC−MS/MS. Different colors represent different groups. Blue, MSA group; red, HSA group. X-axis and Y-axis indicate the first and second principal components (PC1 and PC2), respectively. Score plots for PC1 and PC2 show cohesion within groups and separation between MSA and HAS groups. (**B**) OPLS−DA scatter diagram for HSA and MSA groups. R2X and R2Y represent the explanatory rate of the model to x and y matrices, respectively. (**C**) Volcano plots of the DAMs for HSA vs. MSA. VIP > 1, *p* value < 0.05. Red points, upregulated DAMs; blue points, downregulated DAMs; gray points, metabolites with no significance. (**D**) Top 20 of KEGG enrichment terms for HSA vs. MSA. The *p* value is presented in a color scale, the size of the dots represents DEG number mapped in each pathway. The blue arrows indicate amino acid metabolic pathways, the blue lines indicate organic acid metabolic pathways.

**Figure 6 ijms-24-04062-f006:**
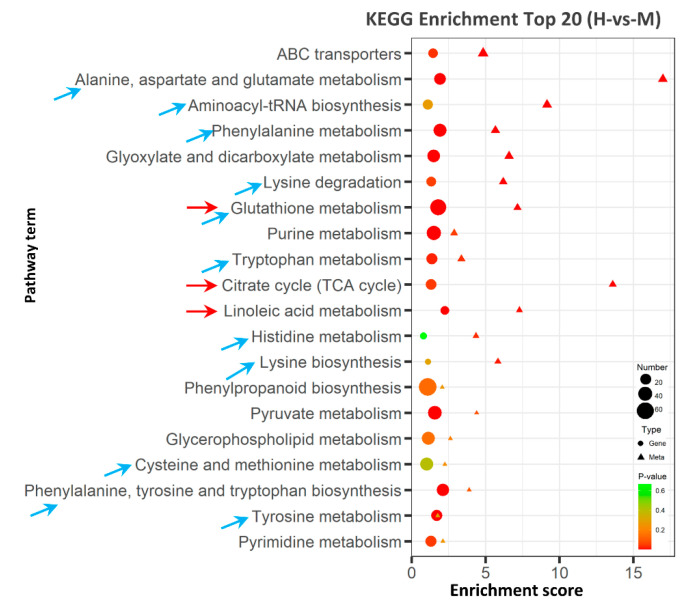
Integrated analysis of KEGG enrichment TOP20 of DEGs and DAMs. The X-axis indicates the enrichment score of the DEGs and DAMs. The *p* value is presented in a color scale, the size of the dots and triangles present DEG and DAM number mapped in each pathway, respectively. The blue arrows indicate amino acid metabolic pathways, and the red arrows indicate the pathways analyzed in detail.

**Figure 7 ijms-24-04062-f007:**
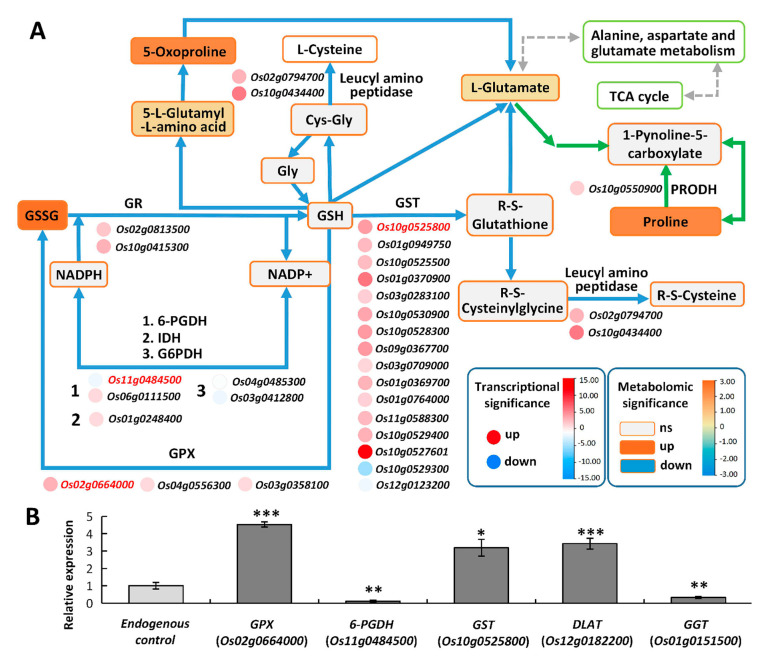
Overview of part of the glutathione metabolism pathway in response to HSA stress. (**A**) The DAMs and related DEGs involved in glutathione metabolism. The rectangles represent the metabolites. The colors of rectangles indicate significances which are presented in a color scale. Gray indicates no significance. The circles indicate DEGs. The colors indicate significances shown in a color scale. The solid lines with arrows indicate directions of the processes. The blue lines indicate processes in the glutathione metabolism pathway, and the green ones indicate processes in arginine and proline metabolism (part process related to proline production). Gray dotted lines indicate connection of other metabolism pathways. The genes highlighted by red were detected using RT−qPCR. (**B**) The gene expression levels were validated via RT−qPCR. Data are presented from three independent experiments performed with three replicates per experiment. *, *p* < 0.05; **, *p* < 0.01; ***, *p* < 0.001; Student’s t−test. Abbreviations: 6-phosphogluconate dehydrogenase (6-PGDH); L-cysteinylglycine (Cys-Gly); gamma-glutamyl transferase (GGT); glucose-6-phosphate dehydrogenase (G6PDH); glycine (Gly); glutathione peroxidase (GPX); glutathione reductase (GR); reduced glutathione (GSH); glutathione disulfide (GSSG); glutathione S-transferase (GST); isocitrate dehydrogenase (IDH).

**Figure 8 ijms-24-04062-f008:**
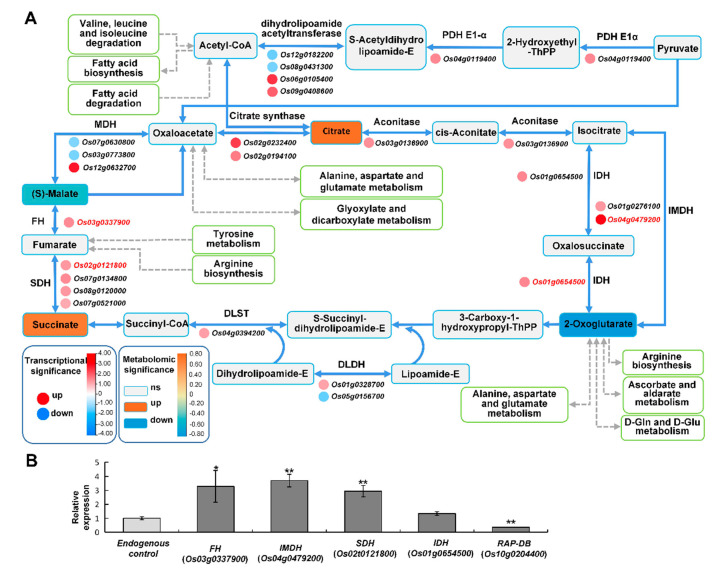
Overview of the TCA cycle in response to HSA stress. (**A**) The DEGs and related DAMs involved in TCA cycle. The meanings of rectangles, circles, colors, and lines are the same as those in Figure 7. The rectangles with the white color and green frame are other metabolic pathways linked to TCA cycle. (**B**) The gene expression levels validated using RT-qPCR. Data are presented from three independent experiments performed with three replicates per experiment. *, *p* < 0.05; **, *p* < 0.01; Student’s t-test. Abbreviations: fumarate hydratase (FH); dihydrolipoamide succinyltransferase (DLST); dihydrolipoamide dehydrogenase (DLDH); isocitrate dehydrogenase (IDH); isopropylmalate dehydrogenase (IMDH); malate dehydrogenase (MDH); pyruvate dehydrogenase (PDH); phosphoenolpyruvate carboxykinase (RAP-DB); succinate dehydrogenase (SDH).

**Figure 9 ijms-24-04062-f009:**
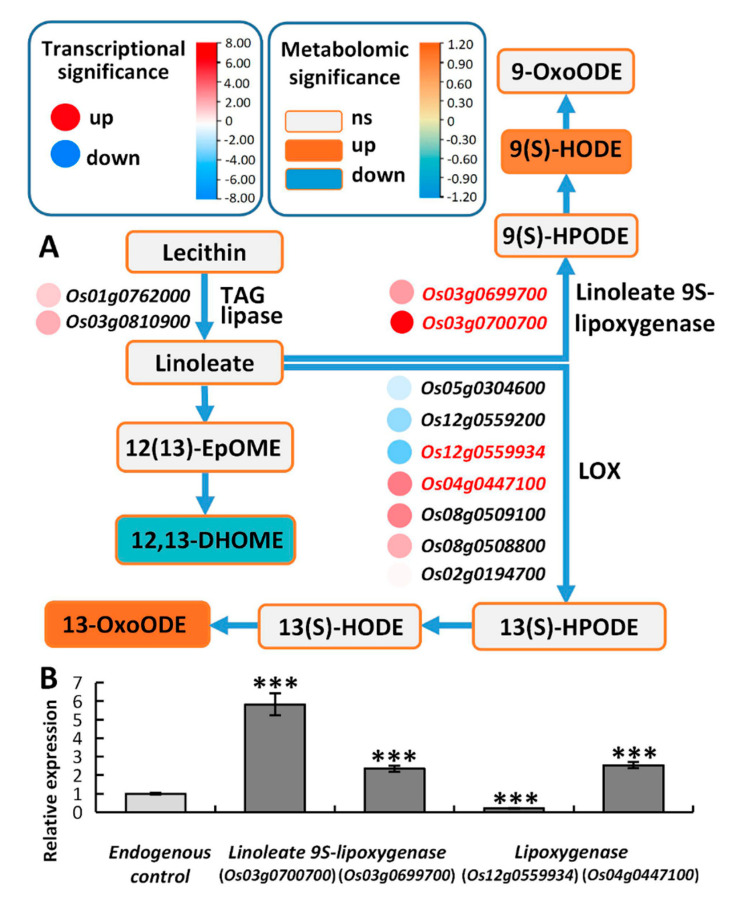
Overview of the linoleic acid metabolism pathway in response to HSA stress. (**A**) The DEGs and related DAMs involved in linoleic acid metabolism. The meanings of rectangles, circles, colors, and lines are the same as those in Figure 7. (**B**) Data are presented from three independent experiments performed with three replicates per experiment. ***, *p* < 0.001; Student’s t-test. Triacylglycerol lipase (TAG lipase).

**Table 1 ijms-24-04062-t001:** Basic nutrient status of the soil.

Sample	pH	Organic Matter (g/kg)	alkali-Hydrolyzable Nitrogen (mg/kg)	Available Phosphorus (mg/kg)	Available Potassium (mg/kg)	Available Zinc (mg/kg)	Salt Contents (g/kg)
MSA	8.41	5.9	34.8	7.6	99.6	1.08	0.29
HSA	9.64	12.8	46.9	10.2	185	0.42	1.24

MSA, mild saline–alkali; HSA, high saline–alkali.

**Table 2 ijms-24-04062-t002:** RNA sequencing data quality.

Sample	Total Raw Reads (Mb)	Total Clean Reads (Mb)	Total Raw Bases (Gb)	Total Clean Bases (Gb)	Valid Bases (%)	Clean Reads Q30 (%)	GC (%)
MSA	50.49	49.81	7.58	7.23	95.4	95.20	49.59
HSA	49.20	48.53	7.38	7.05	95.5	95.12	50.27

MSA, mild saline–alkali; HSA, high saline–alkali.

## Data Availability

Not applicable.

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
