# Peer review of "Integrated Transcriptome and Metabolome Analysis of Rice Leaves Response to High Saline–Alkali Stress"

_ijms, 2023, doi:10.3390/ijms24044062_

Round 1
Reviewer 1 Report
Qian et al. manuscript entitled as “ Integrated Transcriptome and Metabolome Analysis of Rice
Leaves Response to High Saline-alkali Stress” to improve the manuscript follow the following suggestions.
1. Figure 1 A and B should add a scale as a reference.
2. Quantitative real-time PCR verification in the material method should describe the period of the material or the description should be consistent with RNA-Seq.
3. The legend of Figure 2, 3A, 4, 5A-C is not clear.
4. Combined with transcriptome and metabolome, the author clarified that glutathione metabolism, TCA cycle, linoleic acid metabolism, and amino acid and fatty acyl metabolism are important for enhancing saline-alkali tolerance. In this study, whether the author has found important genes and metabolites that may resist long-term saline-alkali stress can be explained in the discussion.
Author Response
- Figure 1 A and B should add a scale as a reference.
Response: Thanks for your suggestion. We have added scale bars as a reference in Figure 1 A and B.
- Quantitative real-time PCR verification in the material method should describe the period of the material or the description should be consistent with RNA-Seq.
Response: Thank you for your suggestion. We have added a description of the material used for quantitative real-time PCR verification. Please see line 545-546 in the revised manuscript.
- The legend of Figure 2, 3A, 4, 5A-C is not clear.
Response: Thank you for your helpful advices. We have optimized the legend of Figure 2, 3A, 4, 5A-C. Please check them in the revised manuscript.
- Combined with transcriptome and metabolome, the author clarified that glutathione metabolism, TCA cycle, linoleic acid metabolism, and amino acid and fatty acyl metabolism are important for enhancing saline-alkali tolerance. In this study, whether the author has found important genes and metabolites that may resist long-term saline-alkali stress can be explained in Discussion.
Response: Thank you for your suggestion. We have added important genes and metabolites that may resist long-term saline-alkali stress in the Discussion. Please see line 413-427 in the revised manuscript.
Reviewer 2 Report
The manuscript "Integrated Transcriptome and Metabolome Analysis of Rice Leaves Response to High Saline-alkali Stress" is well written however, i have follwoing concern with the manuscript.
1. I am curious to know if there are some common SEF between, MSA and HSA group. If yes, those number and name should be highlighted and supplied.
2. Plant pathogen interaction, plant hormone signal transduction genes pathways showed significant contribution during KEGG analysis. Authors here focused only on Glutathionine but ignored rest two pathways for the downstream analysis. Role of glutathionine in the stress tolerance is well known and metabolites related to this pathway essentially will play important role. However, it would be interesting to now how other pathways interacting or cross talking with this pathway. Authors must study these interactions.
3. A high salt tolerant variety was compared in two different stress condition, however, a comparison between non tolerant varieties should be compared with non tolerant varieties (Authors can use some publically available data for that).
4. In metabolomic analysis also, was there some phytohormones/ compounds those expression was incresased (apart from Glutathionine). If yes, those part should be highlighted providing future prespectives.
These comments must be addressed before publication.
Author Response
- I am curious to know if there are some common SEF between, MSA and HSA group. If yes, those number and name should be highlighted and supplied.
Response: Since we don't know what the abbreviation of 'SEF' is, so unfortunately we can't answer this question.
- Plant pathogen interaction, plant hormone signal transduction genes pathways showed significant contribution during KEGG analysis. Authors here focused only on Glutathionine but ignored rest two pathways for the downstream analysis. Role of glutathionine in the stress tolerance is well known and metabolites related to this pathway essentially will play important role. However, it would be interesting to now how other pathways interacting or cross talking with this pathway. Authors must study these interactions.
Response: Thank you for your kind suggestion. To explore the relationship between Glutathione metabolism and Plant pathogen interaction or Plant hormone signal transduction, we used the STRING protein interaction prediction database (https://cn.string-db.org/) to investigate interaction between the DEGs in the pathways. The results is described in line 428-450 in the revised manuscript.
- A high salt tolerant variety was compared in two different stress condition, however, a comparison between non tolerant varieties should be compared with non tolerant varieties (Authors can use some publically available data for that).
Response: Thank you for your suggestion. Previous studies have reported the regulation mechanism of wild rice under salt stress. we analyzed the similarities and differences of the mechanisms between wild rice and Tongxi726. Please see line 451-467 in the revised manuscript.
- In metabolomic analysis also, was there some phytohormones/ compounds those expression was incresased (apart from Glutathionine). If yes, those part should be highlighted providing future prespectives.
Response: Thanks for your suggestion. In this study, a non-targeted metabolomics approach based on HPLC-ESI-MS/MS was employed to identify the DAMs. Unfortunately, there was no phytohormone-like compounds was detected.
Reviewer 3 Report
General comments
The present manuscript investigated the integrated analysis of transcriptome and metabolome to elucidate the effects of long term saline-alkali stress on rice cultivars. e. High saline-alkali stress (pH > 9.5) induced significant changes of gene expression and metabolites, including 9,347 differentially expressed genes (DEGs) and 693 differentially accumulated metabolites (DAMs). The manuscript sounds scientific and hold potential for understanding of saline-alkali tolerant mechanisms in rice and provides references for molecular design breeding of saline-alkali resistant rice. However some points are suggested to improve the overall quality of the manuscript before final publication.
Comments and suggestions for authors:
Introduction: The introduction is too lengthy and fussy. It must highlight the novelty of your research problem with supportive literature. There is a lack of connection between different sub-sections. The authors did not mention why they have chosen rice for this study. The authors must focus on the current status of rice under saline-alkali stress rather than general introduction of salinity and its detrimental effects. It should be rewritten.
Section 2.1:
delete “serious”.
Rectify spacing error” pH 8.41”.
The authors did not mention, at what concentration of salinity, maximum enzymatic activity was attained.
The activity was compared to whom. What was the control in this study?
Section 4.1: Plant Materials and Treatments: delete “were” planted at….
For RNA, extraction, what types of leaves were taken and at what stage?
Section 4.7 and 4.9 must be merges as single section for Statistical analysis.
Figures: All the figures must have high resolution. These should be modified.
Strictly follow the journals pattern; some sections are italics while some others are not. Correct it.
Discussion: It is very confusing. It should be more precise and informative.
Conclusion: It should be precise and highlights only the major findings of the present study.

Author Response
General comments
The present manuscript investigated the integrated analysis of transcriptome and metabolome to elucidate the effects of long term saline-alkali stress on rice cultivars. e. High saline-alkali stress (pH > 9.5) induced significant changes of gene expression and metabolites, including 9,347 differentially expressed genes (DEGs) and 693 differentially accumulated metabolites (DAMs). The manuscript sounds scientific and hold potential for understanding of saline-alkali tolerant mechanisms in rice and provides references for molecular design breeding of saline-alkali resistant rice. However some points are suggested to improve the overall quality of the manuscript before final publication.
Comments and suggestions for authors:
Introduction: The introduction is too lengthy and fussy. It must highlight the novelty of your research problem with supportive literature. There is a lack of connection between different sub-sections. The authors did not mention why they have chosen rice for this study. The authors must focus on the current status of rice under saline-alkali stress rather than general introduction of salinity and its detrimental effects. It should be rewritten.
Response: Thanks for your suggestion. We have rewritten the introduction to make it more concise and organized, increased the current status of research on saline-alkali stress in rice, and pointed out the reasons for selecting rice as the subject of this study. Please see introduction in the revised manuscript.
Section 2.1:
delete “serious”.
Rectify spacing error” pH 8.41”.
The authors did not mention, at what concentration of salinity, maximum enzymatic activity was attained.
Response: Thanks for your suggestion. We have deleted "serious" and rectified spacing error "pH 8.41" in section 2.1. Please see line 104 and 109 in the revised manuscript, respectively. In addition, we have added content showing what saline concentration maximum enzymatic activity. The added description is " Under HSA stress, the MDA content increased significantly (Figure 1C), suggesting that HSA stress induced excessive ROS production, leading to serious membrane lipid peroxidation. In response, the activities of antioxidases, such as SOD, POD, APX and CAT, and proline content in rice leaves increased sharply (Figure 1D, E), indicating that HSA and ROS stress strongly activated the ROS scavenging and osmotic adjustment systems which maintain membrane integrity and cell activities.". Please see line 114-119 in the revised manuscript.
The activity was compared to whom. What was the control in this study?
Response: In this study, we conducted integrated analysis of transcriptome and metabolome to elucidate the effects of differences saline-alkali stress levels on rice. The rice cultivar ‘Tongxi926’ (Oryza sativa), widely planted in Northeast area in China, has a certain saline-alkali tolerance (productive under mild saline-alkali condi-tion). The test paddle field was with pH9.64 and 1.24 g/kg of salt contents (Table 1), which belonged to severe alkali but mild salt conditions, therefore, we designated it as high saline-alkali (HSA) condition. The rice (Oryza sativa) growth and development was severely inhibited in HSA fields, with dwarf, reduced tillering number and many reddish brown leaves, we used it as experimental group. Another paddle field was with pH 8.41 and 0.29 g/kg of salt contents, which belonged to mild saline-alkali (MSA) condition. Since Tongxi926 grew well under this condition, we used it as control condition.
Section 4.1: Plant Materials and Treatments: delete “were” planted at….
Response: Thanks for your suggestion. We have deleted " were " in section 4.1. Please see line 511 in the revised manuscript.
For RNA, extraction, what types of leaves were taken and at what stage?
Response: Thanks for your suggestion. We have added the description of this part as follow: " The rice leaves at booting stage were harvested for RNA Sequencing. For sampling of control rice leaves, the green leaves were randomly collected; for test rice leaves, the reddish-brown leaves were randomly collected (shown in Figure 1B)". Please see line 517-519 in the revised manuscript.
Section 4.7 and 4.9 must be merges as single section for Statistical analysis.
Response: Thanks for your suggestion. We have merged sections 4.7 and 4.9 as single section for statistical analysis. Please see the revised manuscript.
Figures: All the figures must have high resolution. These should be modified.
Response: Thanks for your suggestion. We have optimized the figures to make them clearer for the reader. Please see figures in the revised manuscript.
Strictly follow the journals pattern; some sections are italics while some others are not. Correct it.
Response: Thanks for your suggestion. We carefully reviewed the journal format; and made uniform changes to the article chapters. Please see the revised manuscript.
Discussion: It is very confusing. It should be more precise and informative.
Response: Thanks for your suggestion. We have rewritten the discussion to make it more precise and informative. Please see the revised manuscript
Conclusion: It should be precise and highlights only the major findings of the present study.
Response: Thanks for your suggestion. We have rewritten the conclusion to make it more precise, as follow: " In this study, we focused on the important genes and metabolites involved in the long-term saline-alkali stress in rice leaves by transcriptome and metabolome analysis. A total of 693 DAMs and 9,347 DEGs were identified for HSA-vs-MSA, respectively. The DEGs of transcription factors mainly enriched in bHLH, ERF, MYB, NAC, C2H2, WRKY, and bZIP families, protruding the significance of transcriptional regulation in rice response to HSA stress. Under HSA stress, the contents of amino acids and lipids increased most, suggesting their essential roles on improving rice tolerance to saline-alkali stress. The DEGs and DAMs were most significantly enriched in the pathways of amino acid biosynthesis and metabolism, ABC transporter, glyoxylate and dicarboxylate metabolism, glutathione metabolism, TCA cycle, and linoleic acid metabolism, indicating that these pathways are important for regulation of rice response to saline-alkali stress." Please see conclusion in the revised manuscript.
Reviewer 4 Report
Dear Authors
Please make all changes in the attached PDF file

Author Response
"‰" changed to "%
Response: Thanks for your suggestion. "‰" have been changed to "%". Please see line 33 the revised manuscript.
Please delete it
Response: Thanks for your suggestion. We have deleted "analysis". Please see line 139 the revised manuscript.
Round 2
Reviewer 2 Report
No comments